# Overcoming Interpretability and Accuracy Trade-off in Medical Imaging

**Ivaxi Sheth**[1,2]                              IVAXI-MITESHKUMAR.SHETH.1@ENS.ETSMTL.CA

**Samira Ebrahimi Kahou**[1,2,3]              SAMIRA.EBRAHIMI-KAHOU@ETSMTL.CA

[1] *Mila Quebec AI*

[2] *ÉTS Montréal*

[3] *CIFAR AI Chair*

## Abstract

Neural networks are considered black boxes. Deploying them into the healthcare domain poses a challenge in understanding model behavior beyond the final prediction. There have been recent attempts to establish the trustworthiness of a model. Concept-based models provide insight into the model by introducing a bottleneck layer before the final prediction. They encourage interpretable insights into deep learning models by conditioning the final predictions on intermediate predictions of explainable high-level concepts. However, using concept-based models causes a drop in performance which poses an accuracy vs explainability trade-off. To overcome this challenge we propose coop-CBM, a novel concept-based model. We validate the performance of coop-CBM on diverse dermatology and histopathology images.

**Keywords:** Interpretability, concept-based explanations

## 1. Introduction

With the growing use of Deep Learning (DL) based decision-making, it is of paramount importance that these models are transparent in their decision-making. The field of medical imaging has observed tremendous advancements in recent times with the aid of computer vision algorithms, assisting radiologists and pathologists in accurately diagnosing diseases (Ardila et al., 2019). However, despite its success, the lack of transparency in decision-making by deep learning models remains a concern due to the potential consequences of errors made by such models (Chen et al., 2022). eXplainable AI (XAI) aims to address these concerns by developing techniques that allow us to better understand the reasoning behind the decisions made by AI systems. Post-hoc explanation methods allow visualization to provide insights into the model's behavior and identify which input features are important for prediction (Selvaraju et al., 2017; Kim et al., 2018). However, these methods involve extra probing and are not inherently interpretable models. Intrinsically interpretable models in medical imaging can therefore provide a deeper understanding and confidence to the healthcare professionals in using DL based Computer Aided Diagnostics (CAD) (Borys et al., 2023). Concept learning models have gained popularity in explaining their own predictions while conditioning on human-understandable concepts. Koh et al. (2020) proposed Concept Bottleneck Model that first predicts concepts, and using those concepts, the final label is predicted. CBM although portrays an explainable model, it is at the expense of the lower accuracy of the model. Inaccurate diagnostics are equally as undesirable as opaque black-box models. In this work, we, therefore, propose a novel concept-based architecture, *coop-CBM* that overcomes the trade-off between interpretability and accuracy.

## 2. Proposed Method

Standard classification models digest an image to output a label. Such models might be explained by using activation visualization or post-hoc vectors after training (Zhou et al., 2018). They aren't although innately explainable and require probing. Our model, *coop-CBM*, is a hybrid multi-task model that predicts both labels and explanations.

**Setup** Consider a standard supervised learning setting for a classification task, where models $\mathcal{M}$ are trained on a dataset $\mathcal{D} = \{x_i, y_i\}_{i=1}^{K}$ with $K$ data samples. Standard models aim to predict the true distribution $p_{\mathcal{M}}(y|x)$ from an input $x$. In the *supervised concept-based model* setting, the dataset has additional labeled concepts that can allow supervised concept learning in addition to target learning. The dataset $\mathcal{D} = \{x_i, c_i, y_i\}_{i=1}^{K}$ is the input to concept-based model. The model has prediction at two levels, the first model $\mathcal{G}_{X \to C}$ maps the input image $x$ to concepts $c$ denoted by $p_{\mathcal{G}}(c|x)$, while the second model $\mathcal{F}_{C \to Y}$ maps the concepts $c$ to the label $y$ denoted by $p_{\mathcal{F}}(y|c)$. During inference, such models are particularly advantageous as they allow model editing based on human feedback. If a supervisor observes incorrect concepts related to a label, they can correct the output of $p_{\mathcal{G}}(c|x)$ which effectively changes, often improves, the downstream label prediction $p_{\mathcal{F}}(y|c)$.

Despite being an explainable model, Mahinpei et al. (2021) have shown that concept representations of CBMs result in information leakage which deteriorates predictive performance. Apart from this, another challenge of CBMs for medical images is the lack of fine-grained concept annotations. These concept annotations in originally define the visual aspects of the images. Due to the diversity in images and lack of expert knowledge, it is difficult to acquire them. To overcome the lack of concepts, one can use the meta-data of a patient that often includes both descriptive features of the image such as tumor size, and non-descriptive attributes such as sex. On such datasets, CBMs suffer from poor performance since the concept bank is not sufficiently expressive (Havasi et al., 2022). To overcome this tradeoff between interpretability and predictive performance, we propose *coop-CBM*.

**Coop-CBM** To preserve the standard model's performance, our model, coop-CBM uses a supplementary predictor. Non-bottleneck models that use concepts as auxiliary features have most commonly been used in multi-task setup (Zhou et al., 2018). But such models lose the causal property and thus lose the cause $\to$ effect explanations. Therefore inspired by the literature on multi-task learning, we introduce an additional predictor, $\mathcal{H}_{X \to Y}$ that predicts supplemental label. This additional stream is separate from the concept prediction pipeline. We hypothesize that this supplementary label prediction helps the concept prediction stream to recover model performance in the absence of fine-grained concept labels.

## 3. Results

In this work, we proposed coop-CBM which overcomes the tradeoff of interpretability and accuracy, we evaluate our model against the current variants of CBM. All of our experiments use the Inception V3 (Szegedy et al., 2016) backbone. We consider two classification datasets, TIL (Saltz et al., 2018) and DDI (Daneshjou et al., 2021, 2022) to classify cancer tumors and skin diseases respectively. These two datasets are different in their concept representation. The metadata for TIL includes non-image features such as age and gender along with clinical descriptor terms. There 185 such concepts in TIL. In the case of DDI, only 48 clinical descriptor terms are present.

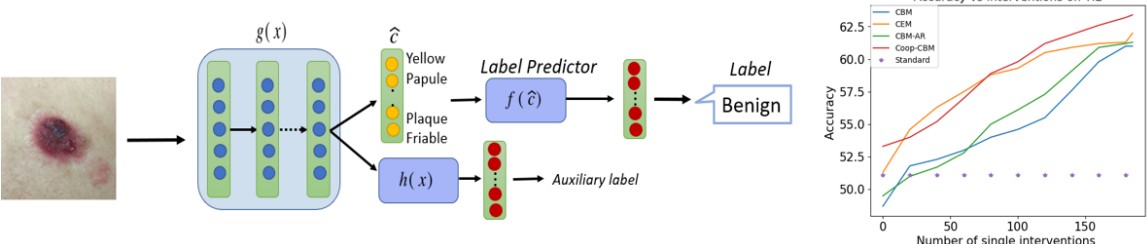

Figure 1: L: Coop-CBM model for DDI data. In addition to predicting concepts in the bottleneck, our model also predicts the supplementary label. The final label is predicted from concepts. R: Performance of different models with interventions on TIL.

**Performance**  To evaluate the performance, here, we are concerned with the final prediction accuracy, i.e. performance of $p_{\mathcal{F}}(y|c)$. From Table 1, we notice our method has the most superior performance in comparison to the baselines on both TIL and DDI datasets. We observe that Concept Bottleneck Models (Koh et al., 2020) observe a big drop in performance in comparison to the Standard model that does not use concepts. Concept Embedding Models (Zarlenga et al.) that build upon the (Koh et al., 2020) by introducing a mixture of concept embedding in the bottleneck layer is just marginally better than the standard model. Finally, Autoregressive CBM (Havasi et al., 2022) also performs comparably to CBM. Coop-CBM improves generalization accuracy even beyond "no concept" models enabling a higher level of explainability without loss of performance.

| Model type | TIL | DDI |
|---|---|---|
| Standard *[No concepts]* | 51.1 | 83.4 |
| CBM (Koh et al., 2020) | 49.0 | 79.9 |
| CEM (Zarlenga et al.) | 51.3 | 83.9 |
| CBM-AR (Havasi et al., 2022) | 49.5 | 80.6 |
| Coop-CBM (ours) | **53.4** | **84.0** |

Table 1: Accuracy of different on TIL and DDI dataset.

**Interventions** An advantage of introducing a bottleneck layer before the final prediction is the ability to perform concept correction during inference (Koh et al., 2020; Sheth et al.). From the perspective of a medical professional, identification of key medical concepts such as skin lesion color may be easier than making a final diagnosis. Therefore if the doctor observes an incorrect concept explanation during test time, they can intervene and alter the concepts often resulting in superior downstream performance (Wang et al., 2022). To quantify the effectiveness of interventions, we compare the accuracy with increasing intervention by choosing concepts randomly and correcting them to ground truth. Figure 1 shows that coop-CBM is highly receptive to concept correction on TIL.

In summary this paper tackles the trade-off between concept related interpretability and predictive performance by proposing multi-task based learning paradigm *coop-CBM*.

**Acknowledgments**

We would like to thank the Digital Research Alliance of Canada for computing resources and CIFAR for research funding. IS acknowledges funding from Mitacs and Imagia Canexia Health.

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
