# OpenReview forum: "Overcoming Interpretability and Accuracy Trade-off in Medical Imaging"
_MIDL.io/2023/Short_Paper_Track — MIDL 2023 Short paper track Poster_

### Official Review · Reviewer_z98K · 2023-04-20
**Well written paper with some technical questions to address**

**Rating:** 7
**Confidence:** 3

**Review:**

Summary: This paper proposes a method that can help with the interpretability and accuracy trade-off for concept-based models. Existing concept-based models are good from an interoperability perspective but compromise accuracy. This paper solves this problem by introducing a separate auxiliary stream of label prediction while still maintaining the main concept branch.

Strengths:

The motivation is clear and the auxiliary stream to increase the predictive power seems to be intuitive.

The method seems to perform well, especially on the TIL dataset.

The paper overall is well-written and easy to understand.

Weaknesses:

It is not very clear how exactly are the auxiliary labels used. A description of what the loss for auxiliary labels looks like would help with a better understanding of the method.

---

### Official Review · Reviewer_ZMB3 · 2023-04-20
**Review for paper 87**

**Rating:** 7
**Confidence:** 4

**Review:**

Interesting idea about balancing the tradeoff between explainability and accuracy.